# Probing the Atomic Structure of Californium by Resonance Ionization Spectroscopy

Felix Weber [1,*], Christoph Emanuel Düllmann [2,3,4], Vadim Gadelshin [1], Nina Kneip [1], Stephan Oberstedt [5], Sebastian Raeder [3,4], Jörg Runke [4], Christoph Mokry [2,3], Petra Thörle-Pospiech [2,3], Dominik Studer [1], Norbert Trautmann [2] and Klaus Wendt [1]

[1] Institut für Physik, Johannes Gutenberg-Universität Mainz, 55099 Mainz, Germany; gadelshin@uni-mainz.de (V.G.); nina.kneip@uni-mainz.de (N.K.); dstuder@uni-mainz.de (D.S.); kwendt@uni-mainz.de (K.W.)

[2] Department Chemie—Standort TRIGA, Johannes Gutenberg-Universität Mainz, 55099 Mainz, Germany; duellmann@uni-mainz.de (C.E.D.); mokry@uni-mainz.de (C.M.); pthoerle@uni-mainz.de (P.T.-P.); ntrautma@uni-mainz.de (N.T.)

[3] Helmholtz-Institut Mainz, 55099 Mainz, Germany; s.raeder@gsi.de

[4] GSI Helmholtzzentrum für Schwerionenforschung GmbH, 64291 Darmstadt, Germany; runke@uni-mainz.de

[5] European Commission, Joint Research Centre (JRC), 2440 Geel, Belgium; stephan.oberstedt@ec.europa.eu

\* Correspondence: wfelix02@uni-mainz.de

**Abstract:** The atomic structure of californium is probed by two-step resonance ionization spectroscopy. Using samples with a total amount of about $2 \times 10^{10}$ Cf atoms (ca. 8.3 pg), ground-state transitions as well as transitions to high-lying Rydberg states and auto-ionizing states above the ionization potential are investigated and the lifetimes of various atomic levels are measured. These investigations lead to the identification of efficient ionization schemes, important for trace analysis and nuclear structure investigations. Most of the measurements are conducted on $^{250}$Cf. In addition, the isotope shift of the isotopic chain $^{249-252}$Cf is measured for one transition. The identification and analysis of Rydberg series enables the determination of the first ionization potential of californium to $E_{IP} = 50,666.76(5)$ cm$^{-1}$. This is about a factor of 20 more precise than the current literature value.

**Keywords:** californium; resonance ionization spectroscopy; ionization potential; atomic structure

## 1. Introduction

The radioactive element californium (Cf) with an atomic number of $Z = 98$ is one of the exotic heavier members of the actinide series. As a rare exception in this part of the nuclear chart, this element features a series of longer-lived isotopes that can be produced in high flux nuclear reactors. Nevertheless, only lower-lying atomic states of californium have been studied so far, while information on the higher-lying atomic states around the first ionization potential (IP) is almost completely missing [1]. Extending the available information is of fundamental relevance for the characterization of the element and will support the identification of efficient photo-ionization schemes, which are a precondition for, e.g., high resolution laser spectroscopy as well as ultra-sensitive trace analysis investigations in the range of the heavier actinides [2]. On top of that, detailed studies of the hyperfine structure and isotope shift will provide further insight into the nuclear structure in this region of the nuclear chart [3,4]. Californium, in particular, is of specific relevance for investigations of nuclei at the deformed sub-shell closure at neutron number $N = 152$, coinciding with $^{250}$Cf. This shell structure has already been predicted by Seaborg in 1989 [5] and has been confirmed by high precision mass measurements [6] as well as decay spectroscopy [7], and updated in [8]. The californium isotopes in the vicinity, i.e., $^{249-252}$Cf, all have half-lives ($T_{1/2}$) in the order of 2 to 1000 years. They provide the only accessible isotope series around $N = 152$, which can be produced in mg quantities

and thus easily be studied. Starting from curium as seed material, the californium isotopes are bred by successive neutron capture and subsequent $\beta^-$-decays in the high flux isotope reactor (HFIR) at the Oak Ridge National Laboratory (ORNL) in Oak Ridge, TN, USA [9]. In particular, the isotope $^{252}$Cf ($T_{1/2} = 2.64$ a) is well recognized for its use as a starter neutron source in nuclear reactors, for medical applications and in the oil industry in well logging applications, based on its properties as a strong neutron emitter [10,11].

Californium was discovered in 1950 by Thompson et al. when it was produced by the irradiation of $^{242}$Cm with 35 MeV helium ions at the Berkeley 60-inch cyclotron [12]. Its chemical behavior was found to be very similar to that of dysprosium, its isoelectronic homologue. First, investigations on the atomic spectrum of californium were performed in 1962 in a spark source [13], followed by the use of electrodeless lamps in the 1970s [14,15]. This work was extended in 1994 and led to a compilation of 136 even and 265 odd parity low-lying levels in neutral californium (Cf I), for which the angular momenta and even the electronic configurations were assigned for most levels [1].

In the current investigations, resonance ionization spectroscopy (RIS) was used as a versatile and highly sensitive technique to study the atomic structure by step-wise excitation and finally photo-ionization of the element of interest [16,17]. Due to the efficient generation and detection of ions, this method is extremely sensitive and can be applied on samples in the pg range with atom numbers well below $10^9$ [18]. The first laser spectroscopic investigations on californium applying RIS were carried out in 1996 on a sample containing about $10^{12}$ atoms of $^{249}$Cf ($T_{1/2} = 351$ a). A resonant two-step laser excitation followed by a third ionization step using another tunable laser was used to determine the IP by electric field ionization, which led to a value of $E_{IP} = 50,665(1)$ cm$^{-1}$ [19]. The analysis of the convergence limit of series of high-lying Rydberg levels is usually a more precise and reliable way to determine the IP of an element, although, in open f-shell elements like the lanthanides or actinides, this can be ambiguous due to the complex atomic spectra and the resulting variety of interactions and level mixings [20–22]. In californium, the complexity of the atomic spectrum around the IP is still unknown, as no studies are reported in this region. In this way, the present studies contribute to the understanding of these complex atomic structures with multiple open shells and acts as experimental support for the theoretical predictions of the entirely unknown atomic systems in the range of the super-heavy elements.

In this work, the investigation of the atomic structure of californium in this high-lying energy region around the first IP is presented. The studies include the search for strong auto-ionizing states (AIS) starting from six different odd-parity low-lying energy levels, which were populated through optical excitation steps from the ground-state. After the identification of efficient ionization schemes, some of these first excited states (FES) were investigated in more detail with respect to their lifetime and the saturation behaviour of the corresponding ground-state transition. The observed Rydbergs series converging to both the ionic ground-state and the lowest-lying excited state in the Cf$^+$-ion were analysed to determine the IP. In addition, the isotope shift for the ground-state transition at 419.91 nm was investigated in the isotopic sequence $^{249-252}$Cf with low resolution as preparation for future work.

## 2. Experimental Setup

A californium sample (isotopic composition: $^{249}$Cf: 25.6%; $^{250}$Cf: 31.4%; $^{251}$Cf: 13.4%; $^{252}$Cf: 29.6% at the time of the measurements) was purchased from Eckert & Ziegler Nuclitec GmbH (E & Z) as 0.1 M nitric acid solution and prepared at the Department of Chemistry's TRIGA Site at Johannes Gutenberg University Mainz (JGU). Two aliquots, each containing in total about $10^{10}$ atoms of $^{249-252}$Cf, were extracted from this solution and each aliquot was pipetted onto a $7 \times 7$ mm$^2$ zirconium foil of 25 µm thickness. After evaporation of the solution, the foils were folded for complete encasement of the samples, and then inserted into an atomizer tube. The RIS measurements were conducted at the RISIKO mass separator of the Institute of Physics at JGU. It offers a high efficiency, which is crucial for spectroscopy

on minuscule samples. The principal layout of the RISIKO apparatus is depicted in Figure 1, while a more detailed description can be found in [23].

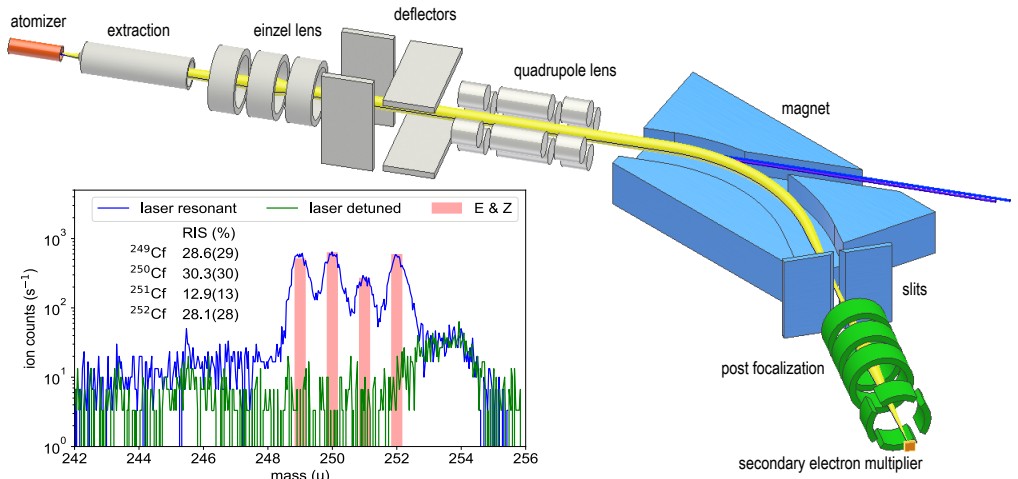

**Figure 1.** Sketch of the RISIKO mass separator with the ion trajectory indicated in yellow and the laser beams in blue. More details on the apparatus are given in [23]. The inset shows the mass spectrum of the californium sample measured with resonant ionization (blue trace) and with first-step laser detuned (green trace). The isotope ratio expected from the certificate of the material is shown in red. More details are given in the text.

The tantalum atomizer tube can be heated resistively to a maximum temperature of about 2000 °C. Already at atomizer temperatures around 800 °C, neutral californium atoms evaporate from the sample and an ion signal from resonant laser ionization is observed. This is in reasonable agreement with the evaporation behavior described in [24]. The ions are accelerated to 30 keV and guided as collimated ion beam through a 60°-sector-field dipole magnet, which separates them by their mass-to-charge ratio. An adjustable separation slit is used to select only a single mass, providing a mass resolution of $M/\Delta M \approx 600$. The mass-selected ions are re-focused by an einzel lens and counted by a secondary electron multiplier. The ionization process is induced by two custom-built pulsed Ti:sapphire lasers, both featuring automatic frequency scanning and tracked intra-cavity phase matching for second harmonic generation [25]. The average output powers range between 200 and 800 mW with a pulse repetition rate of 10 kHz, a pulse length of 30 to 50 ns and a continuous scanning range from 350 to 500 nm. Both Ti:sapphire lasers are individually pumped with up to 18 W power provided by commercial high repetition rate pulsed frequency-doubled Nd:YAG-lasers operating at 532 nm. A pulse generator is used to trigger the pump pulses for optimum synchronization required for efficient ionization. Alternatively, the pulse generator can be used to generate a variable time delay between the excitation steps to probe the lifetime of an excited state.

The inlay of Figure 1 shows the corresponding mass spectrum with all four californium isotopes being clearly visible as obtained by applying a two-step ionization with both lasers on resonance. For comparison, a spectrum was included with the first excitation laser detuned from the resonance. The latter was recorded to ensure that the signal obtained on resonance can be completely ascribed to californium and also in order to determine the remaining background conditions on the individual mass settings. Non-related mass peaks, like the structure at mass 254, show no dependence on the applied laser wavelength. The expected isotopic composition (certified by E & Z) is drawn as red bars and shows good agreement with the pattern obtained by RIS (*cf.* inlay Figure 1). Marginal uncertainties in the values determined by the RIS method can be ascribed to influences of the optical isotope shift, which was not taken into account during these measurement, to the different hyperfine structure patterns of the odd-mass isotopes or finally to minimal statistical

fluctuations in the ion signal. For these reasons, an uncertainty of 10 % of the determined value was adopted for each isotope.

## 3. Laser Spectroscopic Investigations

In the accessible energy range for the first excitation step from 23,000 to 28,720 cm$^{-1}$, 15 energy levels are listed in [1] that can be directly excited by electric dipole transitions, starting from the $5f^{10}7s^2$ $^5I_8$ ground-state. To confirm the completeness of optical ground-state transitions in this range, a frequency scan of the first excitation laser was performed, while keeping the second laser at a fixed wavelength of 360 nm with 400 mW average power to ensure photo-ionization from any resonantly excited level. In this scan, all known energy levels could be confirmed while no additional resonances were found. This measurement was performed using $^{249}$Cf, which allowed a comparison to the data in [1]. Additionally, it created the opportunity to search for particularly broad resonance structures, which might be suitable candidates for future hyperfine structure measurements. The obtained results are in excellent agreement with the previous work. However, transitions that are prominent due to particularly large linewidths were not found in this broadband laser scan. In the next step, six different first excitation steps and the corresponding autoionizing structure have been measured on $^{250}$Cf (T$_{1/2}$ = 13.08 a), which was chosen to avoid possible influences of hyperfine structure in the spectra. The investigated ionization schemes are depicted in Figure 2 and discussed in the following.

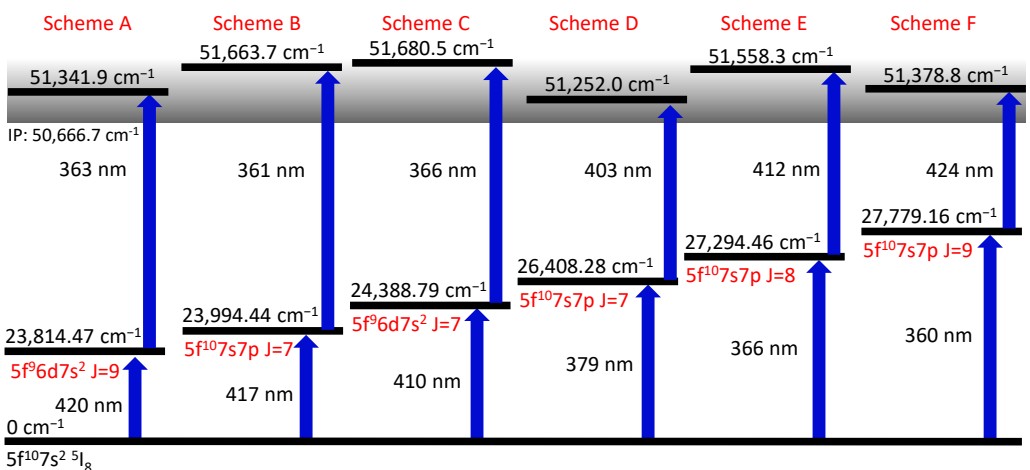

**Figure 2.** Sketch of the ionization schemes identified in this work. All energetic positions refer to $^{250}$Cf. The electron configurations are taken from [1]. For all schemes, the ionization laser was scanned around the IP to search for AIS and Rydberg-levels. The AIS providing the highest ion count rate is given as a second step.

### 3.1. Ionization Schemes

To identify suitable ionization schemes, the laser driving the ionization step was scanned in the region of the IP starting from one of the six FES of Figure 2. Most scans were performed up to about the first excited level of the ion ($E_{IP}$ + 1180.52 cm$^{-1}$ [1]), while scheme C was deliberately extended up to an excitation energy of 52,000 cm$^{-1}$ to map possible structures above. The obtained spectra show many AIS and Rydberg states, as visible in Figure 3. For each excitation scheme (Figure 2), the resonance resulting in the highest ion count rate is given to provide a collection of resonant two-step ionization schemes in californium. Depending on the first step, several AIS might produce similarly intense ion signals. Unfortunately, the ion count rate obtained from the different spectra cannot be compared directly to each other, due to varying ion source conditions. The laser scans for schemes D and E, e.g., were taken on a fresh, newly inserted californium sample and with a higher atomizer temperature, resulting in a higher ion count rate than the measurements of the other schemes. To compensate for that, an enhancement factor

was determined by normalizing the ion signal to a baseline of non-resonant ionization. This allows for deducing the enhancement of the ion signal due to an AIS directly from the spectra shown in Figure 3. Nevertheless, this enhancement is somewhat biased because, for schemes D, E and F, the laser populating the FES could also provide ionization, which leads to an increased non-resonant baseline compared to the other schemes. Therefore, both types of information, i.e., the ion count rate during the scans and the enhancement compared to the non-resonant baseline, are provided in Figure 3. The distinct series of Rydberg states showing up in the spectra are evaluated in Section 3.4.

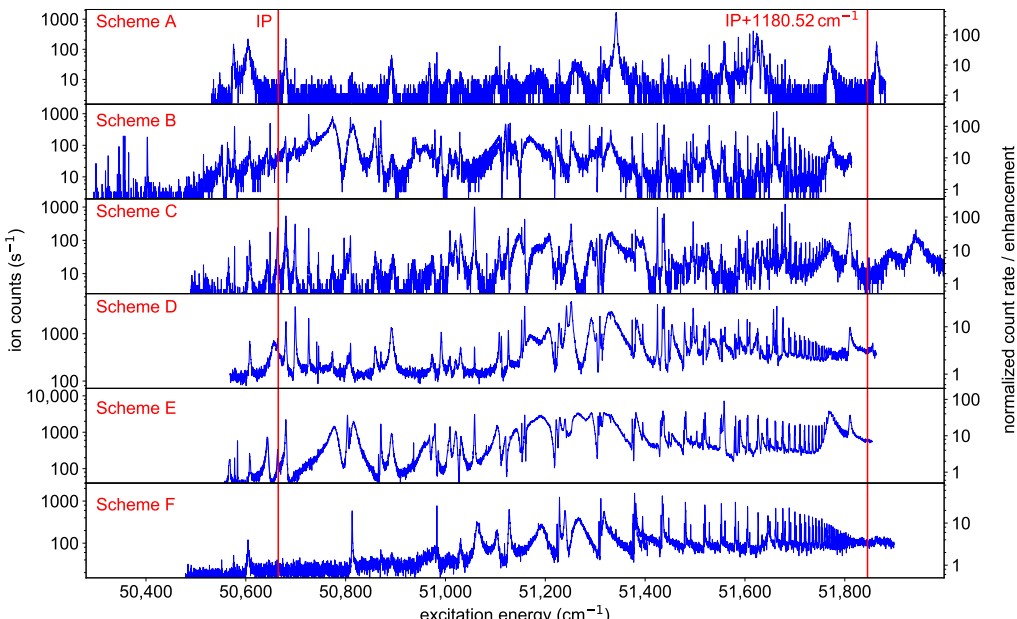

**Figure 3.** Spectra of neutral californium obtained by scanning the ionization laser around the IP starting from either one of the six different FES presented in Figure 2. On the left *y*-axes, the count rate during the measurement is displayed. On the right *y*-axes, an enhancement factor of the ion signal as determined by normalizing the ion signal to the individual baseline of non-resonant ionization is indicated. Distinct Rydberg series show up in nearly all spectra converging to the lowest-lying excited state in the $Cf^+$-ion, located at an excitation energy of 1180.52 cm$^{-1}$ above the IP [1]. The resonance marked with an asterisk is discussed in Section 3.4

### 3.2. Lifetime and Saturation

The lifetime of an excited level is directly linked to the strengths of the involved optical channels for de-excitation. The ground-state transition should be one of the dominant ones in our case. Here, the decay of the FES population is probed by delaying the second, ionizing laser with respect to the first, exciting laser and monitoring the decline of the ion signal with increasing delay-times. The expected curve is a convolution of the nearly Gaussian-shaped laser pulse profile and the exponential decay of the level population [26]. In cases where the first transition is strongly saturated, the ionization rate increases notably when both laser pulses coincide. Then, the convolution cannot describe the complete shape of the decay curve, and an exponential decay is fitted on a subset describing the situation of temporally well separated laser interactions. The obtained results are shown in Figure 4 for schemes A, B and C, where the data in the left panel were measured with 130 mW to 160 mW laser power in the first step and in the right panel with a lower laser power of about 5 mW.

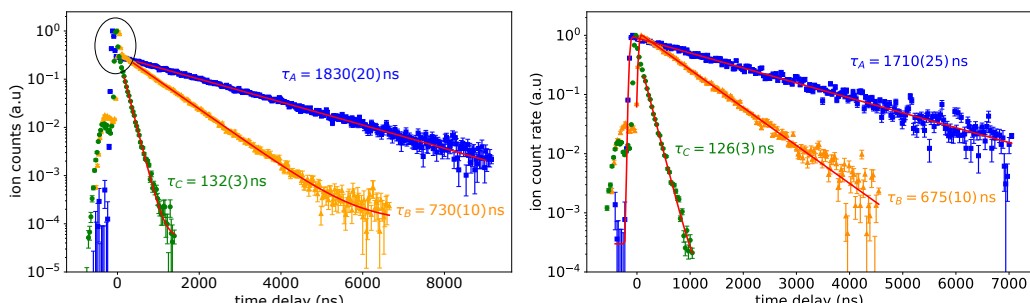

**Figure 4.** Left: lifetime measurement of the FES for schemes A, B and C with laser power of about 130 mW for scheme A, and 160 mW for scheme B and C in the excitation step. The area where both laser pulses overlap is highlighted. Right: same lifetime measurement with a laser power of about 5 mW for all schemes in the excitation step. More details are discussed in the text.

Both graphs show a small peak at the very beginning of the curve at negative time delay values. This is due to a weak second laser pulse following the main pulse of the ionization laser and can be neglected in the analysis. The additional strong peak in coincidence for a time delay of 0 ns (highlighted in the left panel in Figure 4) vanishes in the right panel for schemes A and B, as the laser power was decreased. Therefore, the convolution can be used here to describe the entire data set. In scheme C, the coincidence peak shows up even with reduced laser power, which indicates that the transition was still saturated. Here, the fit is applied to a subset of data only. The obtained lifetime values for schemes A and B differ somewhat between both measurements. The deviation is too large to be explained by statistics only. It is ascribed to varying temperature conditions within the atomizer during a measurement, which are induced by the change of laser power. This could be circumvented by waiting until the ion source conditions had stabilized before starting a new measurement. Future studies will allow for excluding systematic errors in these measurements. Correspondingly, the real value of the lifetime is expected in the range between the values given in Figure 4. It is obvious that the FES from schemes A and B are relatively long-lived.

Efficient ionization requires saturation of the excitation steps. The saturation power $P_{\text{sat}}$ is measured by attenuating the laser power in the first excitation step and monitoring the ion signal. The lasers in both steps were focused on a beam spot size of $\approx 2.5(10)$ mm$^2$ in the atomizer tube. The expected ion signal curve can be expressed by

$$S(P) = C_1 \cdot \frac{1}{(1 + P/P_{\text{sat}})} + C_2 \cdot P + C_3, \tag{1}$$

with laser power $P$, saturation power $P_{\text{sat}}$ and the coefficents for resonant ionization $C_1$, laser-induced background $C_2$, and laser-independent background $C_3$ [27]. In Figure 5, the saturation behavior is shown for the three ground-state transitions of schemes A, B and C.

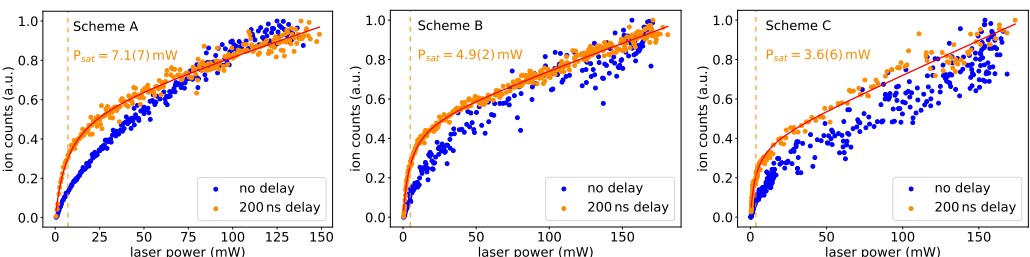

**Figure 5.** Saturation curves for the first excitation steps in schemes A, B and C. For all levels, the curve evolution differs when the ionization laser pulse was delayed. More details are given in the text.

The saturation behaviour differed significantly depending on whether the measurement was made with or without a time delay between excitation and ionization laser

pulses. This was expected, as the strength of the primary peak for coincident laser pulses in the lifetime measurements was depending on the power of the excitation laser. In this case, a realistic saturation power of the transition can only be given for a sufficiently long delay between both pulses. The indicated saturation powers should only be seen as rough estimates, as the size of the laser beam overlapping with the second laser spot was not determined precisely, and power measurements were not performed directly at the point of interaction. Nevertheless, the values demonstrate that all three schemes can easily be saturated in common laser ion sources. As expected, the trend in $P_{sat}$ follows the trend in lifetime, i.e., a longer lifetime corresponds to a higher saturation power.

*3.3. Isotope Shift*

The isotope shift is the frequency difference in an atomic transition between two isotopes with mass numbers $A$ and $A'$, defined as $\delta\nu^{A,A'} = \nu^{A'} - \nu^A$. Its sign provides hints about the configuration of the involved energy levels. A precise determination gives access to nuclear structure parameters such as the change in mean square charge radii and the incorporated nuclear deformation [4]. High-resolution measurements with linewidths of about 100 MHz, including evaluation of the hyperfine structure in $^{249,251}$Cf, were performed by us on the californium samples in a more sophisticated experimental arrangement and will be discussed and published separately by the same authors. A detailed description of the required upgrade of the setup and the resulting high resolution spectroscopic data are beyond the scope of this publication. As preparation for these investigations, the isotope shift of the first step in scheme A was measured here with a spectral linewidth in the order of 2 GHz. The obtained data are shown in Figure 6. The linewidth is mainly determined by the Doppler broadening in the hot atomizer tube and by the laser bandwidth. For this measurement, an etalon was installed in the laser resonator. This reduced the laser bandwidth from the typical values of 5–7 GHz during the wide range scans to typical values of 1–2 GHz. Minor differences in the linewidths for the different isotopes are caused by different laser powers or laser operation conditions.

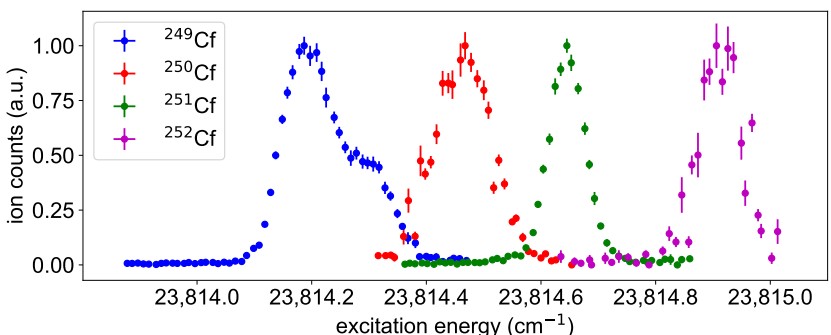

**Figure 6.** Isotope shift of $^{249-252}$Cf for the ground-state transition in scheme A with a linewidth of about 1.5 GHz. The isotope shift has a negative sign, which is in agreement with the assigned configuration of $5f^96d7s^2$ for the upper state.

The transition energy shown in Figure 6 is lower towards lighter isotopes, which indicates that none of the s-electrons in the $5f^{10}7s^2$ ground-state is involved in the transition. This is in agreement with the $5f^96d7s^2$ configuration assigned for the excited state in [1]. The transitions shows a sub-structure for $^{249}$Cf caused by the hyperfine structure. However, due to the nuclear spin of $I = 9/2$ and the high angular momenta of $J = 8$ in the ground-state and $J = 9$ in the FES, 27 individual hyperfine structure components are expected, and therefore an analysis is not possible here. For $^{251}$Cf, the nuclear spin of $I = 1/2$ leads to an expected splitting into just three hyperfine components, but no indication for hyperfine structure is visible in the corresponding peak at a linewidth of about 2 GHz. For these reasons, a quantitative analysis of the isotope shift is not attempted here and reference is given to the paper that will be published separately by the same authors.

### 3.4. Ionization Potential

Distinct Rydberg series converging to the lowest-lying excited state in the $Cf^+$-ion are observed for all schemes except of scheme A (*cf.* Figure 3). In some spectra, e.g., for scheme F, the series split at lower energies, i.e., at a larger distance from the convergence limit, into two or more series. Schemes B and E additionally exhibit series that converge to the ionic ground-state and thus directly allow the determination of the IP. In order to assign the resonances to individual series, all individual resonance peaks were fitted. All parts of the spectra containing Rydberg resonances were scanned twice, once with increasing and once with decreasing wavelength, in order to exclude systematic shifts in the resonances. These may occur due to the scanning speed of the laser and a possible marginal delay in the data acquisition. The mean value obtained from both scanning directions was taken as the energetic position of each peak. The observed shift of all peaks in a series between the two scanning directions was averaged and a common standard deviation was determined, which is taken as the uncertainty of the individual energies for a specific series. The average itself shows how strongly the peak positions are influenced by the scanning procedure and was used for correction if individual resonances could only be observed in one scan. The statistical uncertainty obtained in this way lies between 0.04 and 0.1 $cm^{-1}$, depending on the scanning speed and the statistical quality of the data. Afterwards, the obtained level energies were inserted into the Rydberg–Ritz formula

$$E_n = E_\infty - \frac{R_\mu}{(n - \delta(n))^2} = E_\infty - \frac{R_\mu}{(n^*)^2} \tag{2}$$

to obtain the effective principal quantum number $n^*$. Here, $E_n$ is the energy of the Rydberg-level with the principal quantum number $n$, $E_\infty$ the series limit, $\delta(n)$ the quantum defect and $R_\mu$ the Rydberg constant for finite nuclear mass. The fractional part of the quantum defect $\delta_{frac}$ is in first order constant and independent of $n^*$ for a series, if the convergence limit is chosen properly. Minor variations as a function of $n^*$ can be expressed with the Ritz expansion

$$\delta(n) \approx \delta_0 + \frac{\delta_1}{(n - \delta_0)^2} + \frac{\delta_2}{(n - \delta_0)^4} + (...), \tag{3}$$

where the higher orders are needed to account for alterations in $\delta(n)$ towards lower $n^*$. This is shown in Figure 7 with an assumed convergence limit $E_\infty = 51{,}847.20\,cm^{-1}$ for all schemes. Even for scheme A, Rydberg series can be recognised, although only a few levels could be measured. For this reason, no further analysis was carried out for this scheme. For many of the other schemes, two distinct series show up, which can be assigned to s- (green) and d-series (red), with the latter often exhibiting a significant fine structure splitting for lower $n^*$. In such cases, the dominant series of more prominent peaks in the spectrum was considered for the Rydberg fit.

It is noticeable that the plot for scheme F is much cleaner than, e.g., the plot for scheme D. The latter shows in addition a large perturbation around $n^* \approx 42$, so that the energy levels above this value are not taken into account in the further analysis. Specifically the s-series in scheme F shows a well localized perturbation caused by an interloper for $n^* \approx 22$, which can be described within the multi channel quantum defect theory as an additional contribution to the quantum defect [28]

$$\delta_{pert}(n) = \delta(n) - \frac{1}{\pi} \arctan\left(\frac{\Gamma_I/2}{E_n - E_I}\right). \tag{4}$$

Here $E_I$ and $\Gamma_I$ are the energy and the width of the interloper, respectively. It is also visible that, for scheme F, the s- and the d-series are not truly parallel, which results in slightly different convergence limits. This behaviour could be caused by a long range perturbation located close to the convergence limit, which however cannot be clearly identified here. All energy levels, which were used for the extraction of the IP, are highlighted in Figure 7 by the coloured dots. The schemes considered here, except for scheme C, have an $5f^{10}7s7p$

electron configuration in the first excited state so that s- or d-series can be excited. The fractional quantum defects of ≈0.4 and ≈0.7 in Figure 7 can be assigned to absolute quantum defects of ≈5.4 and ≈3.7 for s- and d-series, respectively, in agreement with theoretical expectations [29]. This allows for assigning the principal quantum number to each Rydberg level and to plot it against its excitation energy. The IP is extracted by fitting Equation (2) to this data as shown in Figure 8 for scheme E as an example.

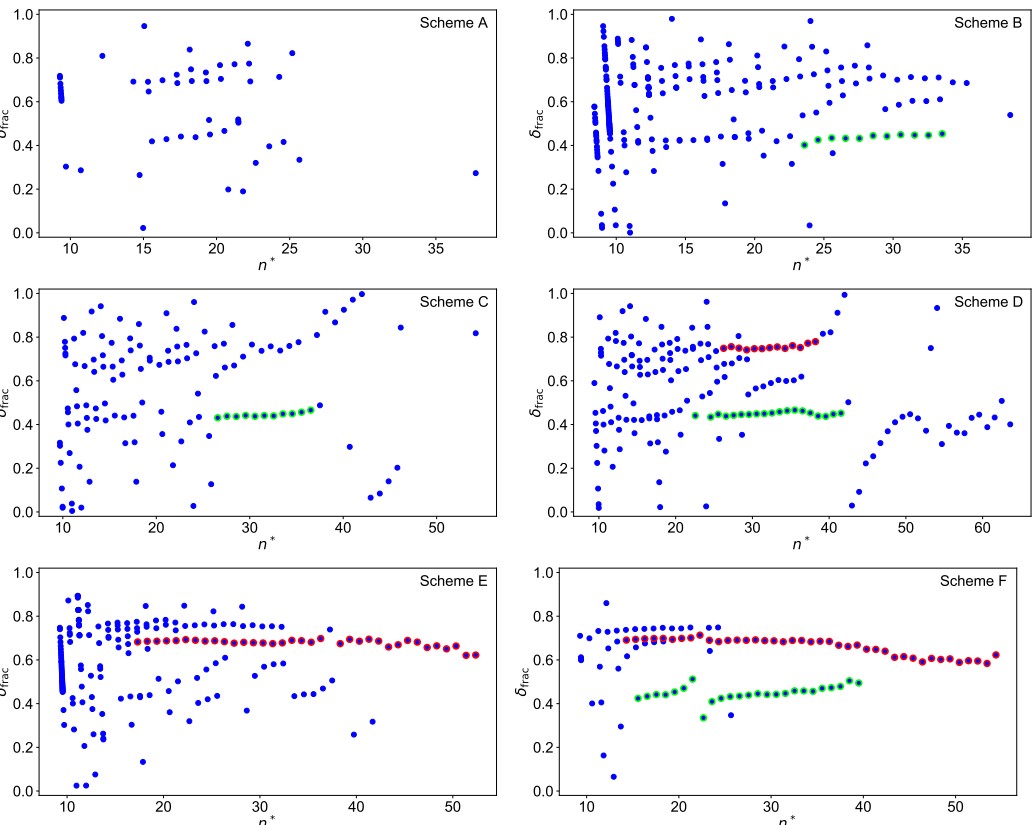

**Figure 7.** Plot of the fractional quantum defect $\delta_{\text{frac}}$ versus the effective quantum number $n^*$ for all schemes with an assumed convergence limit of $E_\infty = 51,847.20$ cm$^{-1}$. Assigned resonances for the Rydberg-fits are highlighted in red for d-series and in green for s-series. Rydberg series converging directly to the IP appear in schemes B and E as nearly vertical lines around $n^* = 10$. More details are given in the text.

Here, two series were identified, one converging directly to the IP and the other one to the lowest-lying excited state in the Cf$^+$-ion. Below the IP, Rydberg-states were observed with principal quantum number from $n = 41$ to $n = 66$. Above, peaks were seen from $n = 21$ to $n = 56$, with a gap for $n = 41$, due to an underlying resonance (cf. the resonance marked with an asterisk, Figure 3, scheme E). To reduce systematic trends in the residuals, an interloper was included in the fit around $n = 26$. In this case, a definite identification was difficult, but the influence on the convergence limit with or without the consideration of an interloper is negligible here anyway. Table 1 summarizes the results for all series that were analysed in a similar way. Only those resonances for which a clear assignment to Rydberg series is possible are considered. If an interloper has to be taken into account as described in Equation (4), this is marked with an asterisk.

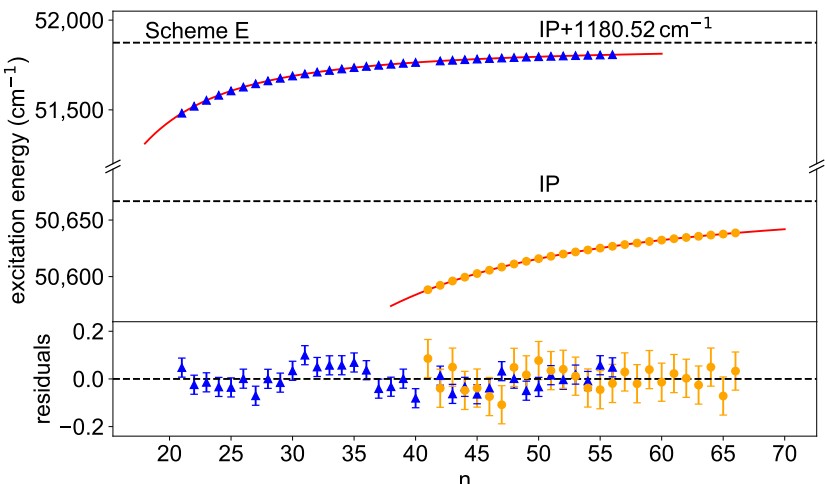

**Figure 8.** Rydberg–Ritz fits for scheme E towards the IP and the lowest-lying excited state in the Cf$^+$-ion. For the latter, an interloper is taken into account at $n = 26$. The residuals show that the model describes the data reasonably well for both cases. The extracted convergence limits are given in Table 1.

**Table 1.** Compilation of the results from the Rydberg-analysis of five different excitation schemes and altogether for nine series; two of them converge directly to the IP of $^{250}$Cf, while the other converge to the lowest-lying excited state in the Cf$^+$-ion, which is located 1180.52 cm$^{-1}$ above the IP [1]. The uncertainty of the mean value is calculated according to the Birge ratio. For series marked with an asterisk, an interloper is considered in the analysis. More details are given in the text.

| $E_1$ (cm$^{-1}$) | $n$ | $\delta_0$ | $\delta_1$ | $E_\infty$ (cm$^{-1}$) | IP (cm$^{-1}$) | $\chi^2_{\text{red}}$ |
|---|---|---|---|---|---|---|
| 23,994.44(1) | 41–70 | 3.64(2) | - | 50,666.75(11) | 50,666.75(11) | 0.99 |
| 23,994.44(1) | 29–39 * | 5.42(4) | - | 51,847.04(34) | 50,666.52(34) | 0.41 |
| 24,388.79(1) | 32–42 | 5.42(1) | - | 51,847.04(16) | 50,666.52(16) | 0.45 |
| 26,408.28(1) | 30–42 | 3.74(2) | - | 51,847.13(24) | 50,666.61(24) | 0.35 |
| 26,408.28(1) | 28, 30–47 | 5.44(1) | - | 51,847.14(16) | 50,666.62(16) | 0.25 |
| 27,294.46(1) | 41–66 | 3.61(2) | - | 50,666.78(13) | 50,666.78(13) | 0.39 |
| 27,294.46(1) | 21–40, 42–56 * | 3.70(4) | −3.6(13) | 51,847.28(8) | 50,666.76(8) | 1.42 |
| 27,779.16(1) | 18–58 * | 3.72(1) | −4.5(6) | 51,847.42(8) | 50,666.90(8) | 0.71 |
| 27,779.16(1) | 21–43 * | 5.46(1) | −10.0(15) | 51,847.17(16) | 50,666.65(16) | 0.68 |
| | | | | weighted mean value | 50,666.76(5) | |

All series contain at least 11 Rydberg levels, with the longest sequence even consisting of 41 levels. The small values of $\chi^2_{\text{red}}$ show that the statistical uncertainties of the individual peaks might be somewhat overestimated. The fitting uncertainties for the convergence limit and the IP stated in Table 1 are increased by a factor of 2.6, resulting from analyzing the Birge ratio [30,31]. This factor leads to a good agreement of all values within their error bars, which would not be the case otherwise. One reason for such a slight deviation could be perturbations due to configuration mixing within some series, which are not accounted for by the Rydberg–Ritz formula. In this case, it is not possible to describe the course completely and the uncertainties can be underestimated, which justifies the procedure. An additional small uncertainty arises for all series converging to the lowest-lying excited state in the Cf$^+$-ion. Its energetic position is only known for $^{249}$Cf, while all measurements here are conducted for $^{250}$Cf. An isotope shift would result in a slight systematic deviation for those two studied series, which converge directly to the IP. Here, this uncertainty is assumed to be negligible. The final value

$$E_{\text{IP}} = 50,666.76(5)\,\text{cm}^{-1}$$

is determined as the weighted mean value of all individual results with the overall statistical uncertainty according to the Birge ratio. This result is in good agreement with the current literature value of $E_{\mathrm{IP}} = 50{,}665(1)$ cm$^{-1}$ [19] and increases the precision by about a factor of 20.

## 4. Conclusions

Extensive studies on the atomic spectrum of neutral californium by laser resonance ionization spectroscopy were carried out and presented here. Based on six different first excitation steps, strong auto-ionizing resonances were identified, resulting in efficient two-step ionization schemes. Three of the first excitation steps and related intermediate levels were investigated in more detail by measuring their lifetimes as well as the saturation behavior of the corresponding ground-state transitions. As the saturation has a strong influence on the ion signal when both laser pulses overlap, a determination of saturation powers was only possible when a temporal delay between the two pulses was applied. The isotope shift of the isotopic chain $^{249-252}$Cf was measured for one ground-state transition with a spectral resolution of about 2 GHz, which was not sufficient for a detailed quantitative analysis of the isotope shift or the hyperfine structure in $^{249,251}$Cf. This will be provided in upcoming high resolution studies the paper that will be published separately by the same authors. Rydberg-series converging to the ionic ground state and to the lowest-lying excited state in the Cf$^+$-ion were analysed to determine the IP of californium to $E_{\mathrm{IP}} = 50{,}666.76(5)$ cm$^{-1}$. This value is in good agreement with the current literature value with an improvement in precision by about a factor of 20.

**Author Contributions:** Formal analysis, F.W.; investigation, F.W., V.G., N.K. and D.S.; project administration, C.E.D. and K.W., resources, C.E.D., S.O., J.R., C.M., P.T.-P. and N.T.; writing—original draft preparation, F.W.; writing—review and editing, F.W., C.E.D., V.G., N.K., S.O., S.R., J.R., C.M., P.T.-P., D.S., N.T. and K.W.; visualization, F.W.; supervision, C.E.D, N.T. and K.W.; funding acquisition, K.W. All authors have read and agreed to the published version of the manuscript.

**Funding:** This research was funded by the Bundesministerium für Bildung und Forschung (BMBF, Germany) under Grant No. 05P18UMCIA.

**Institutional Review Board Statement:** Not applicable.

**Informed Consent Statement:** Not applicable.

**Data Availability Statement:** The data presented in this study are available upon reasonable request from the corresponding author.

**Conflicts of Interest:** The authors declare no conflict of interest.

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
