# Peer review of "Probing the Atomic Structure of Californium by Resonance Ionization Spectroscopy"

_atoms, doi:10.3390/atoms10020051_

Round 1
Reviewer 1 Report
By using a particular variant of laser spectroscopy, i.e. resonant ionization spectroscopy (RIS), the authors have obtained new experimental results on excited energy levels of the articially produced heavy element californium. One of the experimental challenges was that the sample material was available only in very small qunatities (picograms). Nevertheless, the autors of obtained high-quality RIS spectra thanky to the extraordinary sensitivity of their experimental approach. They could measure, e.g., excited-level lifetimes, the isotope shift of a prominently visible transition, and the first ionization energy of Cf, the latter, with an accuracy that is a factor of 20 better than that of the hitherto known value.
The manuscript is well written, guiding the reader in suffiecient depth through all steps of the data analysis. The results are new and certainly of interest to the atomic and nuclear physics communities. I recommend that the paper be published as is.
Author Response
Dear Reviewer,
Thank you very much for the positive feedback on our manuscript on spectroscopic studies of californium. We are pleased that you liked the results and their presentation. Based on the comments and suggestions of the second reviewer, we have implemented some minor changes.
Kind regards
Felix Weber on behalf of all authors
Reviewer 2 Report
Authors have presented extensive study on the atomic spectrum of neutral californium by laser resonance ionization spectroscopy. The manuscript may be accepted for publication with attention to following points;
- Authors should describe briefly the importance of the present study in terms of applications.
- In the discussion of Figure 2, author mentioned that they did measurement for 249Cf for the first step and then they mentioned that they did measurement for 250Cf to avoid influence of hyperfine structure. However, somewhere they mentioned present study is useful to understand hyperfine structure. This point should be cleared.
- In line 200, 201: Author mentioned that they have done sophisticated measurements on 249,251Cf and will be published later, Why so?
- Ion counts observed in Scheme C and F beyond the IP+1180.52 Cm-1 line should be explained in Figure 3.
Author Response
Dear Reviewer,
Thank you for the positive feedback and helpful suggestions for improvement. You can find the responses as well as the improvements to the individual points in the attached file. Additionally, we checked the references for relevance and did minor changes ( replaced old reference 1 by old reference 16; and removed reference 19).
Kind regards
Felix Weber on behalf of all authors
